# Comparison of COVID-19 testing strategies and costs for professional sports teams: A case study of J. League clubs

**Masashi Kamo**[1]*, **Michio Murakami**[2], **Wataru Naito**[1], **Tetsuo Yasutaka**[3], **Seiya Imoto**[4]

1 Research Institute of Science for Safety and Sustainability, National Institute of Advanced Industrial Science and Technology, Tsukuba, Japan, 2 Center for Infectious Disease Education and Research, Osaka University, Osaka, Japan, 3 Geological Survey of Japan, Research Institute for Geo-Resources and Environment, National Institute of Advanced Industrial Science and Technology, Tsukuba, Japan, 4 Division of Health Medical Intelligence, Human Genome Center, Institute of Medical Science, The University of Tokyo, Tokyo, Japan

* masashi-kamo@aist.go.jp

## Abstract

Professional sports teams are entertainment groups that earn income through performances, and they recognize that efforts to prevent the within-team spread of infection that could lead to performance cancellation are important. Infectious disease control involves several costs, some of which are in a trade-off relationship. For example, frequent testing can reduce the spread of infection, but it also leads to increased costs. On the other hand, limiting the number of tests can reduce testing costs, but it increases the revenue loss from players becoming infected and the loss from canceling games. Therefore, a methodology that strikes a reasonable balance between the cost of control measures and the risk of infection is needed. The relationship between infection control measures and the number of infected individuals was investigated through simulations using the susceptible-exposed-infected-recovered (SEIR) model. Two types of testing scenarios as control measures were principally considered: rapid antigen testing or the slower PCR testing (regular-testing scenarios); and regular testing with more frequent, additional testing after the appearance of an infected individual (additional-testing scenarios). Testing fees, revenue loss due to player or staff inactivity as a result of infection, and expenses for postponement or cancelation of matches were considered as costs. Regular antigen testing was found to be more effective than PCR testing in reducing the number of infected individuals and associated costs. There are two main reasons why antigen testing was more efficient: It is less expensive than PCR testing; and the results are available sooner (immediately, versus at least a day of waiting time for the PCR results). This was shown to markedly reduce the number of infected individuals.

## Introduction

In response to the global outbreak of the COVID-19 pandemic in early 2020, many government agencies and private organizations have considered measures to control infectious diseases. As the players and staff involved in sports activities work in close proximity, such

**Data availability statement:** Almost all of the figures can be reproduced using the Excel sheet in S3 Appendix. The source codes of the simulations are available at https://github.com/masashi0209/infection_measure_in_sports_team

**Funding:** The author(s) received no specific funding for this work.

**Competing interests:** T.Y. and W.N. received funding from the Japan Professional Football League. T.Y. and W.N. received grants or contracts from Kashima Antlers F.C. Co., Ltd., Kao Corporation, Yomiuri Giants, Tokyo Dome Co., Ltd., The Yomiuri Shimbun, Keio University, and the Tokyo Metropolitan Government. T.Y. received a grant or contract from Asahi Kasei Corporation. T.Y. and W.N. received consulting fees from the Japan Professional Basketball League, Yomiuri Giants, Nippon Professional Baseball Organization, Tokyo Yakult Swallows, and Mitsubishi Research Institute, Inc. T.Y. received consulting fees from Suntory Holdings Limited. W.N. received consulting fees from the Japan Professional Football League. M.K., M.M., T.Y., W.N., and S.I. attended the New Coronavirus Countermeasures Liaison Council, jointly established by the Nippon Professional Baseball Organization and the Japan Professional Football League, as experts without remuneration. T.Y. and W.N. were advisors to the Japan National Stadium. T.Y. and W.N. are advisors to the Japan Professional Football League

infections are more likely to occur than in the general population [1], and therefore players and staff are required to take infection-control measures such as hand washing, maintaining physical distance, wearing masks, avoiding activities with a high risk of infection, and receiving vaccinations. In addition to these measures, regular testing to detect and quarantine individuals with asymptomatic infections is considered important to minimize the spread of infection [2]. Professional sports teams, of course, earn their incomes by holding events. The spread of infection among teams results in various costs, including reduced attendance due to star players being absent because of infection, and loss of profits due to the cancellation of games because of large-scale infections. Infectious disease control is therefore considered to be a very important task. Japanese professional sports groups, such as those in the professional baseball and professional football leagues, conduct more frequent regular PCR testing for every player and stuff than the general population—typically every 2 weeks to 1 month [3,4].

However, the current testing protocols have not been quantitatively evaluated to determine their effectiveness, raising questions about their actual impact on infection control. While it is reasonable to assume that more frequent testing is more effective in controlling infectious disease (e.g., in the case of COVID-19, that two PCR tests per week is more effective than one), more frequent testing may result in higher costs.

As for-profit entities, professional sports organizations naturally seek to reduce testing costs. Reducing the number of tests will lower costs, but it may increase the likelihood of infectious disease outbreaks within an organization. To achieve a balance between cost savings and infection control, antigen testing with rapid results is being considered as a feasible alternative. Although antigen testing is less costly than PCR testing, and it is relatively easy to increase its frequency, it is also less sensitive than PCR testing in detecting infected individuals [5], and this may increase the number of infections. In the case of professional sports teams, in addition to the trade-off between lower testing costs and increased number of infections, the costs of postponing or canceling matches should be considered in discussions of optimal testing strategies. Studies have proposed effective testing strategies for the general population [6] and for a semi-closed small population [7]. One study has shown that frequent testing using lower-sensitivity methods have been shown to be effective in controlling infectious diseases [8]. However, despite these studies, there is a lack of research specifically addressing the dynamics and economic considerations of infection control within professional sports teams.

This study aimed to quantitatively assess the cost-effectiveness of different testing strategies for professional sports teams, focusing on the trade-offs between infection control, testing frequency, and the associated costs. We analyzed the infectious disease dynamics of COVID-19 in a relatively small sports team by using a model constructed in a previous study [7]. We counted the number of infected individuals, the number of tests, and the probability of event postponement or cancelation that resulted from a large number of infections, given a variety of different testing strategies. We then integrated the quantitative items as monetary costs and assessed the effectiveness of the respective testing strategies.

## Materials and methods

A model of the infectious disease dynamics of COVID-19 was analyzed for the wild-type and Omicron variants under various testing scenarios. Through a primarily online survey, we reviewed the published infection-control measures of a number of professional sports teams. As the information provided by the Japan Professional Football League (J.League) was most useful, we investigated the number of members per team and the testing protocol of the J.League, as an example. The methodology developed in this study can, however, be applied to other sports teams or groups, provided that the appropriate information is available.

## Model setup

The infection dynamics of COVID-19 are represented by a compartment model with continuous time. We divide the states of individuals in the population into susceptible (S), exposed (E), infected (I), and recovered (R), with the infected state further divided into four sub-states: presymptomatic on day 1 ($P_1$), presymptomatic on day 2 ($P_2$), infected with symptoms (Is), and infected without symptoms (Ia). $P_1$, $P_2$, Is, and Ia all indicate infection and are all detectable by testing. State E are not infectious and is not detectable by testing. We simulate the population dynamics, which start with one infected individual and continue until the infection stops. During a short period of time, mutation of the virus can be ignored, and hence we assume that the immunity of R individuals is perfect and they are not infected again. These states are represented in the model as italicized variables (i.e., number of S individuals is $S$, number E individuals is $E$, etc.).

An S individual becomes an E individual at the rate of $\beta S(P_1 + P_2 + Ia + Is)$, where $\beta$ is the rate of infection per day. An E individual becomes $P_1$ at rate $\sigma$ (average duration of the E state is $1/\sigma$ days); a $P_1$ individual becomes $P_2$ at rate 1 (average duration: 1 day); and a $P_2$ individual becomes either Ia or Is at rate 1 (average duration: 1 day). The partition ratio where $P_2$ becomes Is is $\eta$ ($0 \leq \eta \leq 1$), and the ratio where $P_2$ becomes Ia is ($1-\eta$). Both Ia and Is become R at rate $\gamma$ (average duration: $1/\gamma$ days). We considered both the wild-type and Omicron variants, with the difference between the variants being $\sigma$ (3 for the wild type and 1 for Omicron). The transitions among these states are shown in Fig 1.

## Mathematical equations and simulation detail

The dynamics of these states are mathematically described as

$$\frac{dS}{dt} = -\beta S(P_1 + P_2 + Ia + Is)/N,$$

$$\frac{dE}{dt} = \beta S(P_1 + P_2 + Ia + Is)/N - \sigma E,$$

$$\frac{dP_1}{dt} = \sigma E - P_1$$

$$\frac{dP_2}{dt} = P_1 - P_2$$

| | | | |
|---|---|---|---|
| Wild-type | 5 days ($\sigma = 3$) | 7 days | |
| Omicron | 3 days ($\sigma = 1$) | 7 days | |

**Fig 1. Diagram of transitions among the states of individuals.** The average duration of E, $P_1$, and $P_2$ was 5 days for the wild type and 3 days for the Omicron variant. The other epidemic parameters were common to the two variants. (See the Model Parameters section below).

$$\frac{dIa}{dt} = (1-\eta)P_2 - \gamma Ia,$$

$$\frac{dIs}{dt} = \eta P_2 - \gamma Is,$$

$$\frac{dR}{dt} = \gamma(Ia + Is), \tag{1}$$

where $N$ is the total number of team individuals involved in group activities. The value of $N$ depends on the sports type [baseball, football (i.e., soccer), or basketball)]. In the case of the J.League, 30 players (https://www.jleague.jp/club/sapporo/day/#player) and 20 staff is typical (based on an interview with J.League), and hence we set $N$ at 50.

This continuous-time model is solved with an agent-based model in which the number of individuals is discretized. The model is designed as follows. The probability ($\Delta S$) that an S individual becomes E within a short time interval ($\Delta t$) is

$$\Delta S = -\beta(P_1 + P_1 + Ia + Is)/N \times \Delta t.$$

We draw a uniform random number, $\xi$, where ($0 < \xi < 1$), and if

$$\xi < \Delta S,$$

then the S individual becomes E; otherwise they remain as S. Conducting the same operation for all individuals yields the state transitions of the population, within a short time interval, $\Delta t$. We set the value of $\Delta t$ at 0.01 (i.e., performing this computation 100 times yields the 1-day dynamics). The infectious disease dynamics begin with one infected individual and continue until all infected individuals have recovered or been removed from the population. A Monte Carlo simulation was performed by doing this calculation 10,000 times to obtain average values for the various measurements.

## Model parameters

The initial conditions for the model are $I(0) = 1$ and $S(0) = N - 1$, with two COVID-19 variants being considered. The common parameters—the rate at which $P_1$ becomes $P_2$, that at which $P_2$ becomes Ia or Is, and that at which Ia and Is become R ($\gamma$)—are, respectively, 1, 1 and $1/7$ [9]. The partition ratio for Is and Ia ($\eta$) is 0.54 (Is:Ia = $\eta : 1-\eta$) [10]. The ratio at which E becomes $P_1$ ($\sigma$) is 1/3 for the wild type [9], and 1 for Omicron, based on [11], which showed that the average total duration of E, $P_1$, and $P_2$ was 3 days (and as $P_1$ and $P_2$ are of 1 day's duration, the average duration of E is 1).

The basic reproductive number ($R_0$) of Omicron is 9 [12]. Given that sports populations typically take stricter measures (including behavioral restrictions) than does the general public against infectious diseases, we can expect that the effective reproduction number would be lower in the case of our model; however, insufficient information was available to estimate the actual value. We therefore decided to investigate two cases, one with an $R_0$ of 5 and the other with an $R_0$ of 2.5. We term these values $R_0$ throughout our paper.

The basic reproductive number is equal to "infection rate $\times$ duration of states with infection ability," that is

$$R_0 = \beta \times \left(2 + \frac{1}{\gamma}\right).$$

The average duration $(2 + 1/\gamma)$ of states with infection ability is 9 days (1 day for $P_1$, 1 day for $P_2$, and 7 days for Is or Ia). From this equation, the rate of infection is

$$\beta = R_0 / 9.$$

## Testing scenarios

The infected individuals in the population are quarantined after a positive symptom check (by visual inspection of the symptoms and temperature measurement) or detection by testing. The symptom check is conducted daily, implying that, on any given day, all the individuals determined to be in the Is state are quarantined. The daily symptom check is assumed to be cost free.

On the basis of the testing results, individuals in the $P_1$, $P_2$, and Ia states are removed from the population. More frequent testing allows for faster detection of infected individuals but leads to higher costs, and this trade-off raises the question of the optimal testing frequency. The J.League initially conducted 2-weekly PCR testing; however, since August 2022, twice-weekly antigen testing with more rapid results (hereafter referred to as antigen testing) has been conducted. We simulated the infection dynamics under both of these testing scenarios, as well as under weekly testing scenarios. In the simulation, we considered the days of a week (7 days) as a cycle, with Saturday as Day 0 (the day on which a match is played). For the 2-weekly and weekly testing scenarios, the testing is conducted on Friday (Day 6), and for the twice-weekly scenario, on Tuesday (Day 3) and Friday (Day 6). The day of the week on which the first infection of an individual occurred was chosen randomly, and the simulation continued until all infected individuals were either removed from the population or recovered, resulting in zero cases in the population.

In the case of PCR testing, the reading time (i.e., the waiting time from the time the test is performed to the time the results are obtained) was 3 days when the major type of COVID-19 was the wild type, and 1 day when the major type was Omicron. The reading time here includes not only the time it takes to obtain the test results in laboratories, but also the time it takes to collect the sample, send it to the laboratory, and return the results to the club. The reading time in the simulation was set at 3 days for the wild type, with two reading-time cases (1 day and 3 days) for Omicron. In the case of antigen testing, the reading time was set at 0 (i.e., the test result was obtained immediately). During the reading time, players and staff were not quarantined or otherwise restricted.

The sensitivities of the testing were the same as in our previous study [7]. The PCR test sensitivities for individuals in States $P_1$, $P_2$, Is, and Ia were 33%, 62%, 80%, and 80%, respectively [13]. In the case of the antigen test sensitivity, three corresponding values were assumed, each relative to the sensitivity of the PCR test: 35%, 50%, and 70%, respectively. Thus, for example, if the sensitivity of the antigen test is 50%, the absolute sensitivities for the individuals in States $P_1$, $P_2$, and both Is and Ia are 16.5%, 31%, and 40%, respectively. Hereafter, we always express the antigen test sensitivity in terms of its sensitivity relative to PCR testing. Among J.League players and staff during the Omicron epidemic, the sensitivity of antigen testing relative to that of PCR testing was reported to be 0.63 (95% CI: 0.54 to 0.72), independent of the number of days between exposure and testing [14]. The sensitivities assumed in this analysis are therefore close to those observed in the J.League.

As the number of simulations was too great if all possible combinations of sensitivity, testing type, and testing frequency in the regular testing were included, a number of representative testing scenarios were selected (Table 1).

**Table 1. Simulation scenarios: regular testing.** Ten thousand Monte-Carlo iterations of each scenario were conducted, with $R_0$ = 2.5 and 5.0. The scenario codes consisted of 4 terms, the first indicating the variant (W for wild type and O for Omicron), the second the type of test (P for PCR and A for antigen testing), the third the frequency of testing per week (05 for 0.5 times [i.e., 1 time for every 2 weeks], 1 for 1 time, and 2 for 2 times), and the fourth depending on the test type: for PCR testing the reading time (3d for 3 days and 1d for 1 day), and for antigen testing the test sensitivity relative to PCR testing (e.g., 35 indicates an antigen test sensitivity of 35% of the PCR test sensitivity). In the case of scenarios involving no testing, N appears as the second, third, and fourth terms (i.e., W-N-N-N for the wild type and O-N-N-N for Omicron).

| Scenario code | Test type | Sensitivity relative to PCR testing (%) | Reading time (days) | Variant |
|---|---|---|---|---|
| W-N-N-N | No testing, daily symptom check only | – | – | Wild type |
| W-P-05w-3d | 2-weekly PCR | – | 3 | Wild type |
| W-P-1w-3d | Weekly PCR | – | 3 | Wild type |
| W-A-2w-35 | Antigen | 35 | 0 | Wild type |
| W-A-2w-50 | Antigen | 50 | 0 | Wild type |
| W-A-2w-70 | Antigen | 70 | 0 | Wild type |
| O-N-N-N | No testing, daily symptom check only | – | – | Omicron |
| O-P-05w-3d | 2-weekly PCR | – | 3 | Omicron |
| O-P-05w-1d | 2-weekly PCR | – | 1 | Omicron |
| O-P-1w-3d | Weekly PCR | – | 3 | Omicron |
| O-P-1w-1d | Weekly PCR | – | 1 | Omicron |
| O-A-2w-35 | Antigen | 35 | 0 | Omicron |
| O-A-2w-50 | Antigen | 50 | 0 | Omicron |
| O-A-2w-70 | Antigen | 70 | 0 | Omicron |

**Table 2. Simulation scenarios: additional testing.** Again, the scenario codes consisted of 4 terms, the first term indicating the variant (O for Omicron), the second the test type (A for antigen testing and P for PCR testing), the third the test frequency for the additional tests (1d for daily and 2d for every second day), and the fourth the reading time (0d for immediate results and 1d for a 1-day reading time). In the case of PCR testing, immediate test results are unfeasible, but the reading time was nonetheless investigated to reveal the effect of this factor.

| Scenario code | Test type | Reading time (days) |
|---|---|---|
| O-A-1d-0d | Daily antigen | 0 |
| O-P-1d-1d | Daily PCR | 1 |
| O-P-1d-0d | Daily PCR | 0 |
| O-A-2d-0d | Alternate-day antigen | 0 |
| O-P-2d-1d | Alternate-day PCR | 1 |
| O-P-2d-0d | Alternate-day PCR | 0 |

In addition to the regular-testing scenarios, we considered scenarios with additional testing (termed "additional-testing scenarios"). In these latter scenarios, if an infected individual was found, regular testing was suspended, and more frequent testing was performed. The test types were antigen tests and PCR. Test frequencies were considered to be daily and every second day. The PCR reading time was set to 1 day. In addition, for reference, a PCR reading time of 0 days, which is somewhat unrealistic, was considered. Considering all of these combinations of test type, test frequency, and reading time would be too difficult, so we conducted simulations with some typical combinations, which are summarized in Table 2. In the additional-testing scenarios, given the need for a rapid response, we assumed that the turnaround time for receiving the PCR test results was much shorter than in the regular-testing scenario. The PCR test sensitivity was equal to that in the regular-testing scenarios, and in these additional-testing scenarios only the 50% antigen sensitivity was simulated. Again, players and staff were allowed to act without restrictions during the reading time for the PCR test results. In addition to a 1-day reading

time, a 0-day PCR reading time was simulated, to examine the impact of waiting time on the infectious disease dynamics for the Omicron variant dynamics.

## Measurement items

Simulations were conducted from the first infected individual appeared and there were no other infected individuals in the population (due to recovery) or in quarantine (due to daily symptom checking or testing). The main measurement items in the simulations were:

1. Number of infected individuals (excluding the first)

2. Number of days from when the first infected individual appears in a population until the population is free of infected individual through recovery or quarantine

3. Number of infected individuals involved in a match

4. Probability that a match was postponed or canceled owing to mass infection (definition below)

5. Number of antigen tests until all infected individuals recovered or were quarantined, equaling the total of:

    5.1 Number of tests in the regular testing

    5.2 Number of tests in the additional testing

6. Number of PCR tests until all infected individuals recovered or were quarantined, equaling the total of:

    6.1 Number of tests in the regular testing

    6.2 Number of tests in the additional testing.

In the actual testing protocols of some sports teams, to confirm the infection, PCR testing is also conducted on individuals who are quarantined as a result of a positive symptom check and a positive antigen test. Therefore, we also incorporated this type of testing in our simulation (the total number of PCR tests included the number of such confirmatory tests), but in this case the test sensitivity was set at 100%.

In all the scenarios, we conducted 10,000 Monte-Carlo iterations and calculated the average values. Among the other items counted were the numbers of infected individuals detected by antigen testing, PCR testing, and daily symptom checks, and the number of susceptible individuals at the end of the infection (a state in which there are no more infected individuals in a population due to recovery or isolation). For the averages of all the measurement items, see supporting information S1 Table.

Postponement or cancelation of a match occurred when the total number of individuals determined (by testing or daily symptom checks) to be infected reached five or more within a single week; this condition was termed "mass infection." In one run of the Monte-Carlo iteration, mass infection could occur twice (e.g., five or more infected individuals appeared for 2 weeks in a row), but if it did so, only one occurrence was counted. The number of runs (out of 10,000 iterations) in which mass infection occurred at least once was also counted, and the average value was calculated as the probability of mass infection occurring.

## Cost of infection and cost of testing protocol

The total cost of the infection and the testing protocol (*TC*) was expressed as the sum of (1) the cost of regular testing (*RC*) and (2) the cost of revenue loss due to inactivity of the infected individual (*AC*) until the infection ended.

The *RC* (yen/day) is derived by

$$RC = \left( Av \times na + Pv \times np \right) \times N,$$

where *Av* is the unit price of antigen testing (yen/times), *na* is the number of antigen tests per day (times/day), *Pv* is the unit price of PCR testing (yen/times), *np* is the number of PCR tests per day (times/day), and *N* is the number of individuals in the sports team population, including staff (*N* = 50 throughout this paper).

When antigen testing was not conducted at a medical institution but was instead self-conducted, the actual unit price of the test was found to be 800–2,000 yen (https://toamit.jp/ and Amazon.co.jp); it was set at 1,000 yen in the simulation. When covered by insurance, the unit price of PCR testing was 13,500–18,000 yen until FY2021, but it is currently (FY2022) 7,000 yen (https://www.mhlw.go.jp/content/12404000/000863590.pdf (in Japanese)), and a web-search revealed that the unit price without insurance is 2,000 yen to 15,000 yen (https:// covid-kensa.com/, https://yuushikai.or.jp/pcr/, https://wakigeka.com/news/outpatient_n/ entry-150.html(in Japanese)). On the basis of this information, the unit price of PCR testing was set at 10,000 yen.

The probability that an infected individual will appear in a given sports team population on a given day depends on the rate of incidence among the general public. Let $P_0$, then, be the probability that an infected individual will appear on a given day. The probability that at least one other infected individual will appear in a sports team population with *N* individuals ($p_g$) is

$$p_g = 1 - \left( 1 - P_0 \right)^N.$$

When $P_0$ is small, $p_g$ can be approximated as the probability that one other infected individual will appear in the team per day.

The cost *AC* depends on the degree of the outbreak. The cost of infection is the sum of:

1. The total number of tests conducted until all infected individuals have recovered or are quarantined.

2. The revenue loss due to the quarantine of infected players or staff (*Iv*).

3. The cost of postponement or cancelation of matches as a result of mass infection (*Cv*).

The revenue lost (*Iv*) due to quarantining was based on the loss of 10,000 yen/day for an average worker in Japan (the precise value is 10,233 yen, but here it was set at 10,000 yen) (https:// www.mhlw.go.jp/toukei/itiran/roudou/chingin/kouzou/z2020/index.html (in Japanese)). Assuming a standard quarantine period of 14 days, *Iv* was calculated to be 140,000 yen/person. We also computed the case where the revenue loss per day was 10 times that for average workers, as the value of professional players or staff is typically much greater than that of an average worker (see S1 Appendix). For professional athletes, who are in higher-pressure environments than usual, other costs should also be considered, such as the potential impact of long-term COVID and other complications, and the impact of testing protocols on the mental health of players and staff, but these were not taken into account in our study. The cost due to postponement or cancelation of matches (*Cv*) was calculated on the basis of the total cost of stadium rental, lodging and transportation for visiting club players, and match-related operations (security, *TV* coverage, etc.). This cost varies greatly depending on the timing of the cancelation and the size of the stadium and audience. It was estimated to range from 1 to 20 million yen or more; for the purposes of the simulation, we set the cost *Cv* at 3 million yen. The costs incurred as a result of the death of infected players and staff was not considered,

because the risk of death by COVID-19 is low among such individuals, who are generally healthy, and no reports of such deaths have been made. The cost of infection treatment was also not considered, because (as of August 2022) all such costs were covered by Japanese national health insurance.

$AC$ (yen/day) is computed as

$$AC = \left(Pv \times Np' + Av \times Na' + Cv \times Pc + Iv \times Nri\right) \times pg,$$

where $Np'$ is the number of PCR tests in the additional testing, $Na'$ is the number of antigen tests in the additional testing, $Pc$ is the probability that postponement and cancelation of matches will occur, and $Nri$ is the number of infected individuals detected.

Finally, $TC$ (yen/day) is calculated as

$$TC = RC + AC.$$

We computed $TC$ with $P_0 = 10^{-5}$ and $10^{-3}$ for the regular-testing scenarios in Table 1. For the scenarios involving no regular testing (O-N-N-N in Table 1), twice-weekly regular antigen testing with a sensitivity of 50% of PCR (O-A-2w-50 in Table 1), and additional testing (the Table 2 scenarios), the costs were computed as a function of $P_0$ (over the range of $0 < P_0 < 10^{-3}$). For the values used to derive $AC$, see supporting information (S1 Table). Fig 2 summarizes the simulation and the calculation of costs.

## Results

### Effects of regular testing

Fig 3 shows (A) the number of infected individuals (excluding the first such individual), (B) the number of infected individuals involved in matches (per match), (C) the number of days it takes

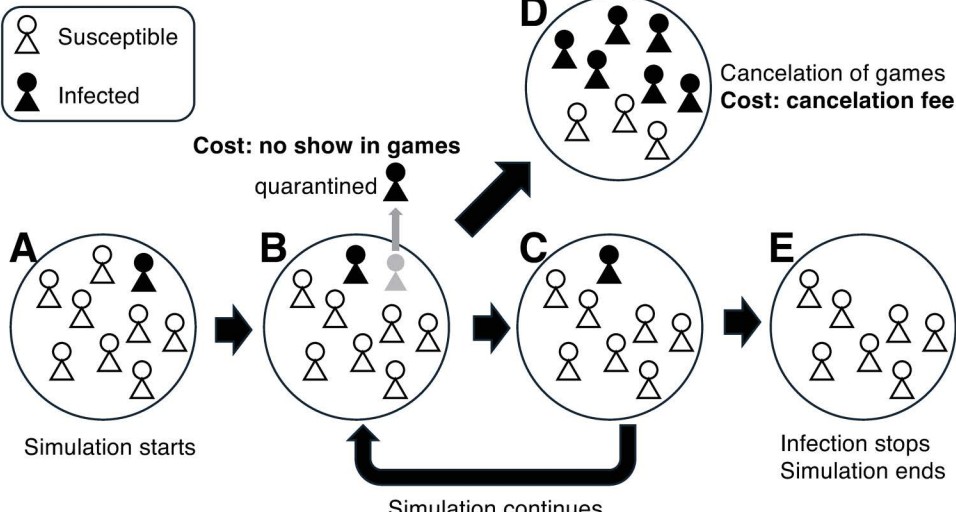

**Fig 2. Summary of the simulation and the cost calculation.** (A) Simulation starts with one infected individual. (B) Infection spreads. If infected individuals are found, the individuals are quarantined. The revenue loss is counted as a cost. (C) Infected individuals remain. The simulation goes back to (B) and continues. (D) If the number of infected individuals is larger than a threshold number, games are cancelled. The cost of the cancelation is counted. (E) If there are no more infected individuals, the simulation ends. In addition to the specified costs, the cost of the testing is counted. The cost depends on the test frequency and the test type (antigen vs. PCR testing). More frequent testing leads to a high isolation rate of infected individuals but costs more.

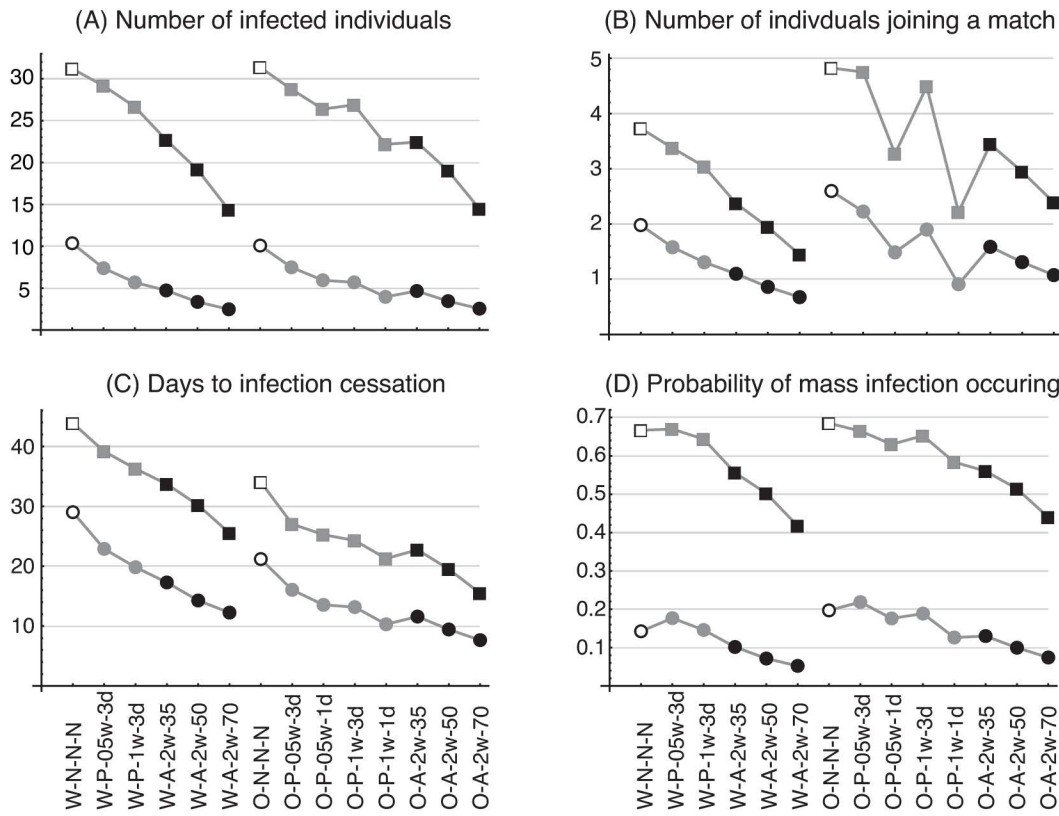

**Fig 3. Average values of four measurement items.** The results for $R_0 = 2.5$ are shown as circles and those for $R_0 = 5.0$ as rectangles. Gray symbols represent PCR testing, black symbols antigen testing, and open symbols no testing.

for the infection to end, and (D) the probability of occurrence of mass infection. The horizontal axis represents the testing scenario (Table 1) and the vertical axis the item measured; open symbols represent no testing, closed gray symbols are PCR testing, and closed black symbols are antigen testing. Rectangles show the results with $R_0 = 5.0$ and circles show the results with $R_0 = 2.5$. These results are categorized into four groups (two variants × two $R_0$), with results belonging to the same category connected by a line (used only to illustrate the respective categories).

Fig 3A shows that, for both $R_0 = 5.0$ and 2.5, twice-weekly antigen testing (black symbols) is more effective at minimizing the number of people infected than no testing (open symbols), 2-weekly PCR testing, or weekly PCR testing (gray symbols), regardless of the sensitivity of the antigen testing. The only exception is that the number of infected individuals in scenario O-P-1w-1d (weekly PCR testing for Omicron with a 1-day reading time) is less than the number of infected individuals in scenario O-A-2w-35 (twice-weekly antigen testing with 35% sensitivity). Fig 3A also shows that the number of infected individuals is the same if the testing scenario and $R_0$ are the same (i.e., scenario pairs such as W-P-05w-3d and O-P-05w-3d, W-A-2w-35 and O-A-2w-35, etc.). It can be shown theoretically that the number of infected individuals is the same for these pairs. (For details, see the section "Effective reproductive number ($Re$) under testing scenarios" and S3 Appendix.) In the case of PCR testing, the reading time affected the number of infected individuals. In PCR testing for Omicron with $R_0 = 5.0$, the number of infected individuals was lower for 2-weekly PCR testing with a reading time of 1 day (scenario O-P-05w-1d) than for weekly PCR testing with a reading time of 3 days (scenario O-P-05w-3d).

Fig 3B shows the numbers of infected players and staff involved in matches. Here again, the number is lower in the case of antigen testing than with no testing (scenarios W-N-N-N, O-N-N-N) or with PCR testing with reading time of 3 days (scenarios W-P-05w-3d, W-P-1w-3d, W-P-05w-3d, and O-P-05w-3d); hence, antigen testing is shown to be more effective than both PCR testing and no testing. The number is also less in scenarios O-P-05w-1d (2-weekly PCR testing with a reading time of 1 day) and O-P-1w-1d (weekly PCR testing with a reading time of 1 day) than in the scenarios with a reading time of 3 days (O-P-05w-3d and O-P-1w-3d), implying that the reading time is an important factor in the infection dynamics. This number is generally higher for Omicron than for the wild-type variant.

Fig 3C shows the number of days until the cessation of infection. The scenarios with twice-weekly antigen testing typically show a shorter period; in the only exception, the period in scenario O-P-1w-1d is less than that in O-A-2w-35. The probability that a mass infection will occur (Fig 3D) is also less with twice-weekly antigen testing than with PCR testing, suggesting that this antigen testing is more effective in COVID-19 control than both 2-weekly and weekly PCR testing. The probability that a mass infection will occur is generally higher with Omicron than with the wild type.

Fig 4 shows the numbers of PCR tests (Fig 4A) and antigen tests (Fig 4B) conducted before the cessation of infection. The number of tests is naturally higher in the weekly testing scenario (W-P-1w-3d) than in the 2-weekly scenario (W-P-05w-3d), because the test frequency in the former is double that of the latter. The number of PCR tests in scenario W-P-1w-3d, however, is not twice as high as in scenario W-P-05w-3d (206 vs. 122 with $R_0$ = 5.0), because the number of infected individuals and the number of days to the cessation of infection are both less in the former scenario (cf. Figs 3A and 3C). As the testing cost is simply proportional to the number of tests, a separate investigation is required to determine whether the increase in the testing cost in scenario W-P-1w-3d is worth the smaller number of infections and days to the cessation of infection. (See the section, "Cost of infection control.") As may be expected, the number of antigen tests decreases as the test sensitivity increases (Fig 4B). In all cases, there are fewer tests in the Omicron scenario than in the wild-type scenario. (Note that the

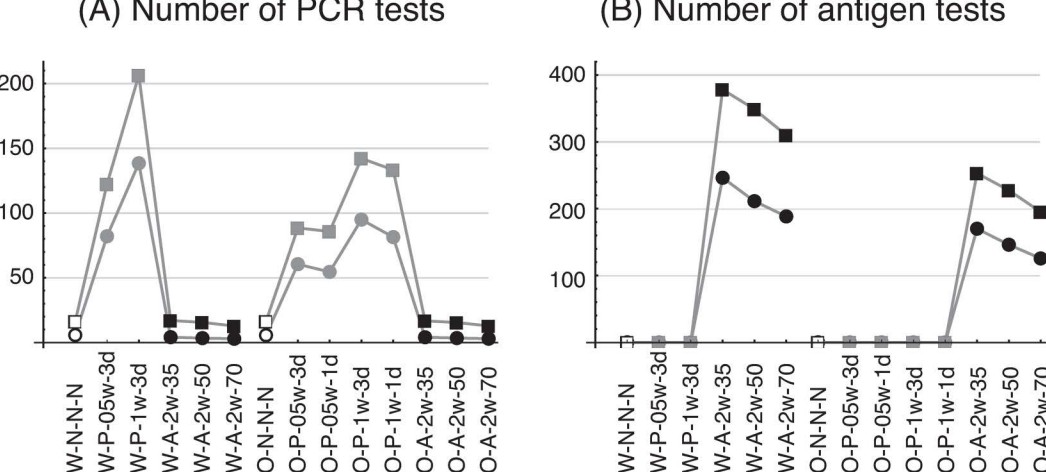

**Fig 4. Number of tests from the time an infected individual appears in the population until the cessation of infection.** (A) PCR testing, (B) antigen testing. Plot symbols and labels are as in Fig 3. In the antigen-testing series (scenario codes W-A- and O-A), the number of PCR tests is not zero. This is because J.League conducted PCR tests to confirm cases that tested positive in the antigen tests. The cost-effectiveness calculation also incorporates the cost of these PCR tests.

number of infected individuals is the same in the respective testing scenarios.) The smaller number of tests in the Omicron scenario is due to the smaller number of days to the cessation of infection.

Overall, implementation of antigen tests twice a week was effective in reducing the risk of infection. The number of tests calculated in the model simulation was used in the cost calculations introduced in "Cost of infection control."

### Effects of additional testing

We also investigated scenarios in which regular testing was conducted until the first infected individual was detected, and more frequent testing was then conducted after this detection. Twice-a-week antigen testing with 50% sensitivity was adopted for the regular-testing evaluation. Only Omicron scenarios were considered in the additional-testing evaluation. The results are shown in Fig 5. Scenarios O-A-1d-0d, O-P-1d-1d, and O-P-1d-0d respectively involved daily antigen testing, PCR testing with a reading time of 1 day, and PCR testing with a reading time of 0 days. Scenarios O-A-2d-0d, O-P-2d-1d, and O-P-2d-0d were similar but with alternate-day testing. In addition to these scenarios, two scenarios of regular testing from Figs 3 and 4 are shown (on the extreme right of each Fig 5 panel) to compare the results of regular and additional testing. Again, the results of PCR testing are shown in gray, those of antigen testing in black, those with $R_0 = 2.5$ in circles, and those with $R_0 = 5.0$ in rectangles.

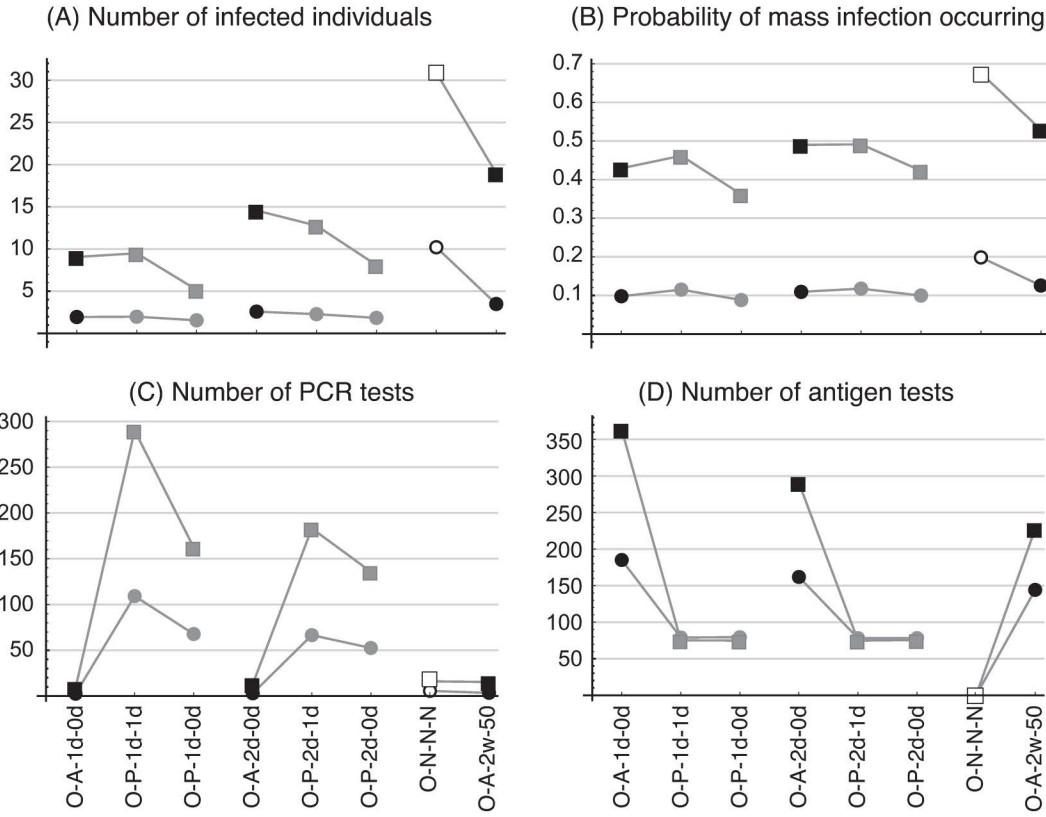

**Fig 5. Results of additional testing.** Gray symbols show the results of PCR testing and black symbols the results of antigen testing; circles show the results for $R_0 = 2.5$ and rectangles the results for $R_0 = 5.0$. In each panel, the two results on the extreme right are for regular testing (from Figs 3 and 4, with open symbols showing the results for Omicron with no testing, and black symbols the results for Omicron with twice-weekly antigen testing at 50% relative PCR sensitivity).

Fig 5A shows the number of infected individuals. If $R_0$ is the same, this number is less with twice-weekly antigen testing (O-A-2w-50) than in the no-regular-testing scenario (O-N-N-N) and the regular-testing scenario. Similarly, the probability of mass infection is less in the additional-testing scenario (Fig 5B). When the frequency of tests in the additional testing decreases (from scenarios O-A-1d-0d and O-P-1d-1d with daily testing to scenarios O-A-2d-0d and O-P-2d-1d with alternate-day testing), the number of infected individuals and the probability of mass infection increases. These results indicate that increasing the testing frequency is an effective infection control measure. As shown in Figs 5C and 5D, the number of tests increases in the scenarios with additional testing, leading to increased costs for the testing protocol. Among the daily additional-testing scenarios (O-A-1d-0d, O-P-1d-1d, and O-P-1d-0d), the number of infected individuals was higher in the scenario for PCR testing with a reading time of 1 day (O-P-1d-1d) than in the scenario for antigen testing (O-A-1d-0d), although the antigen test sensitivity was less than that of the PCR test (50% of the PCR sensitivity).

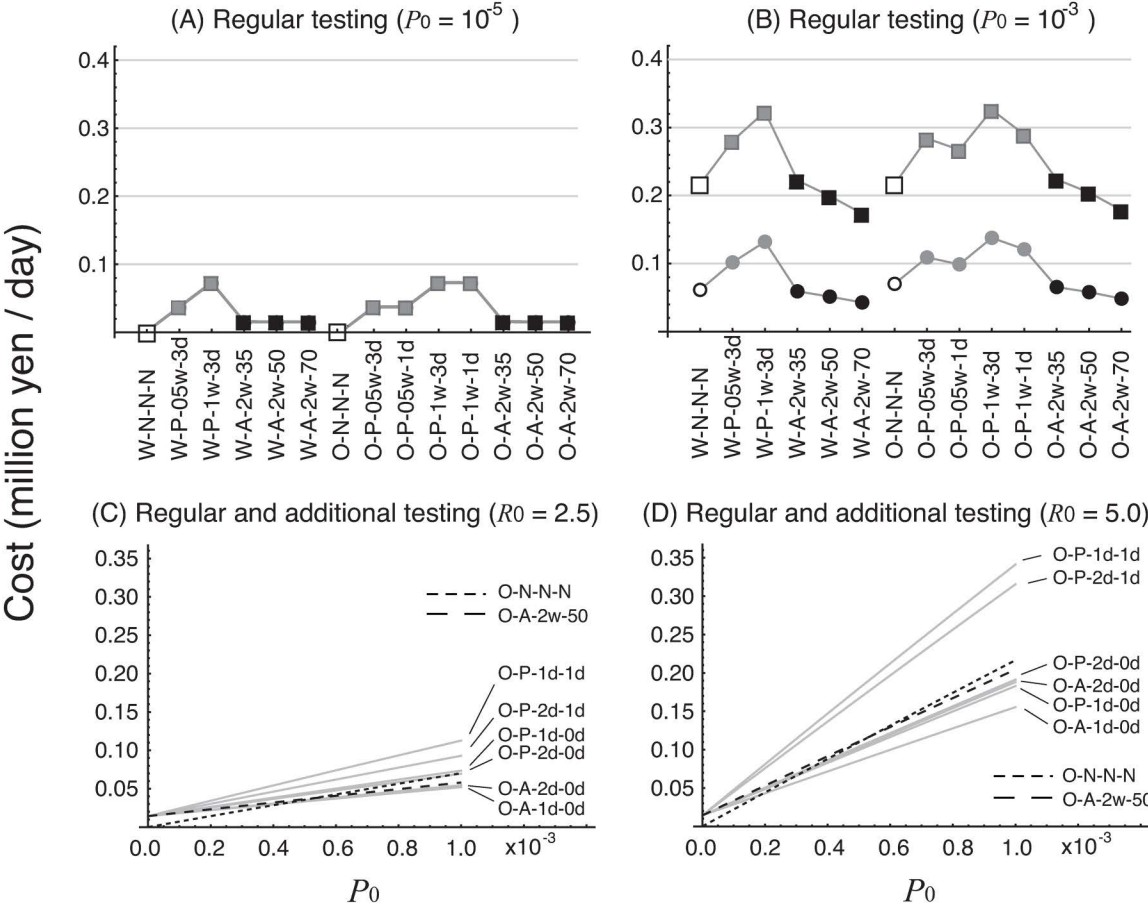

**Fig 6. Costs of testing scenarios.** (A) regular-testing scenarios with $P_0 = 10^{-5}$, (B) regular-testing scenarios with $P_0 = 10^{-3}$, (C) regular and additional-testing scenarios with $R_0 = 2.5$, and (D) regular and additional-testing scenarios with $R_0 = 5.0$. In A and B, the circles represent the cost with $R_0 = 2.5$, and the boxes represent the cost with $R_0 = 5.0$. The two lines (for the results of $R_0 = 2.5$ and 5.0) over-lap in Panel A because the respective values are very similar. With the exception of scenarios O-N-N-N (no regular testing, fine black broken line) and O-A-2w-50 (twice-weekly antigen testing with 50% sensitivity, coarse black broken line), the order of the scenarios regarding costs remains the same, irrespective of $P_0$.

Overall, this section shows the trade-off between reduced risk of infection and increased cost associated with an increased frequency of testing. The cost of implementing testing is discussed below in the "Cost of infection control" section on the basis of these results.

## Cost of infection control

The costs of infection control in the regular-testing scenarios are shown in Fig 6A and 6B. When $P_0 = 10^{-5}$, the difference between the cost with $R_0 = 2.5$ and with $R_0 = 5.0$ was small (e.g., in scenario O-A-2w-50, the cost was 15,000 yen/day at $R_0 = 2.5$ and 16,000 yen/day at $R_0 = 5.0$). In contrast, when $P_0 = 10^{-3}$, the difference in cost was large: 2.36 to 4.04 times higher when $R_0 = 5.0$ than when $R_0 = 2.5$ (e.g., in scenario O-A-2w-50, the cost was 58,000 yen/day at $R_0 = 2.5$ and 200,000 yen/day at $R_0 = 5.0$). When $P_0 = 10^{-5}$, the ratio of $RC$ to $TC$ in scenario O-A-2w-50 was 97% at $R_0 = 2.5$ and 88% at $R_0 = 5.0$; when $P_0 = 10^{-3}$, the ratio was 25% at $R_0 = 2.5$ and 7% at $R_0 = 5.0$. Irrespective of $P_0$ and $R_0$, the costs of the respective twice-weekly antigen testing scenarios (W-A-2w-35, W-A-2w-50, W-A-2w-70, O-A-2w-35, O-A-2w-50, and O-A-2w-70) were less than the costs of the 2-weekly and weekly PCR testing scenarios (e.g., compared with the cost of scenario O-P-05w-1d, the percentage cost of scenario O-A-2w-50 was 41% with $P_0 = 10^{-5}$ and $R_0 = 2.5$, 43% with $P_0 = 10^{-5}$ and $R_0 = 5.0$, 59% with $P_0 = 10^{-3}$ and $R_0 = 2.5$, and 76% with $P_0 = 10^{-3}$ and $R_0 = 5.0$). The cost of the scenarios with no regular testing (W-N-N-N and O-N-N-N) was less than with any other testing scenario when $P_0 = 10^{-5}$ and almost equal to the cost of twice-weekly antigen testing when $P_0 = 10^{-3}$.

The costs of the additional-testing scenarios are shown in Fig 6C and 6D. This cost increased with increasing $P_0$. At $P_0 = 10^{-5}$, the respective costs of the additional-testing scenarios, relative to the cost of twice-weekly regular antigen testing with 50% PCR sensitivity (O-A-2w-50), were 1.00 to 1.03 with $R_0 = 2.5$ and 0.97 to 1.05 with $R_0 = 5.0$. At $P_0 = 10^{-3}$, the relative costs were 0.90 to 1.77 with $R_0 = 2.5$ and 0.77 to 1.43 with $R_0 = 5.0$. Comparison of the twice-weekly 50% sensitivity antigen-testing scenario (O-A-2w-50) with the additional-testing scenarios revealed that, when $R_0 = 2.5$, the costs were lower for O-A-2d-0d (alternate-day antigen testing) and O-A-1d-0d (daily antigen testing), in order of decreasing cost. When $R_0 = 5.0$, the cost was lowest for scenario O-A-1d-0d (daily antigen testing), followed in order by O-P-1d-0d (daily PCR testing with a reading time of 0 days) and O-A-2d-0 (alternate-day additional antigen testing). When $P_0$ was low, the cost of the no-regular-testing scenario (O-N-N-N) was the lowest among all the testing scenarios; however, the cost of this scenario rose sharply compared with those of other testing scenarios, and it exceeded that of scenario O-A-1d-0d (daily additional antigen testing) for $P_0$ greater than $4.4 \times 10^{-4}$ when $R_0 = 2.5$ and for $P_0$ greater than $1.9 \times 10^{-4}$ when $R_0 = 5.0$.

The costs assuming a 10-fold increase in revenue loss due to the inactivity of infected players or staff are shown in supporting information S2 (Fig A1). The following two results were uniformly observed: 1) the cost of the twice-weekly antigen-testing scenario was less than that of the 2-weekly and weekly PCR testing scenarios; and 2) the costs of all the additional antigen-testing scenarios were less than that of any regular-testing scenario. The cost of scenario O-A-1d-0d (daily antigen additional testing) was less than that of scenario O-N-N-N (no regular testing) with $P_0$ greater than $6.4 \times 10^{-5}$ when $R_0 = 2.5$ and with $P_0$ greater than $2.8 \times 10^{-5}$ when $R_0 = 5.0$.

This section shows that, for additional testing, the cost of performing daily antigen tests was low. For regular testing, a low cost was obtained in the case of no tests or antigen tests twice a week, depending on $P_0$.

## Effective reproductive number (*Re*) under infection-control countermeasures

The total number of infected individuals at the time of cessation of infection (final epidemic size: FES) is determined by the basic reproductive ratio ($R_0$) of the infectious disease [15]. This implies that $R_0$ is known if FES is known. By taking countermeasures (quarantine of infected individuals detected by testing or by daily symptom checks), the actual reproductive number is reduced. The reduced reproductive number, which we term the effective reproductive number (*Re*), is known if the FES under the countermeasures is known. Therefore, we first simulate the infection dynamics with no countermeasures (i.e., no daily symptom check or testing) to obtain the FES values corresponding to various $R_0$ values (see S3 Appendix). We then simulate the infection dynamics under a specific testing scenario (e.g., regular antigen testing) to determine the FES value in this scenario. If we then select the FES value closest to the FES with no countermeasures, this will indicate the reproductive number *Re* under this testing scenario.

Fig 7A shows the *Re* values for the regular-testing scenarios. The open symbols indicate scenarios without testing but with a daily symptom check. In the case of the wild type, *Re* is 2.62 when $R_0$ = 5.0 (open rectangles) and 1.44 when $R_0$ = 2.5 (open circles). In the case of Omicron, *Re* is 2.63 when $R_0$ = 5.0 and 1.43 when $R_0$ = 2.5. These *Re* values differ slightly between the Omicron and wild-type variants (e.g., 1.43 vs. 1.44 when $R_0$ = 2.5), but this is simply an error due to the finite nature of the simulations. The results imply that a daily symptom check only has the effect of halving the reproductive number, but *Re* is further reduced when testing is also conducted. The testing scenarios W-A-2w-50 and O-A-2w-50 (regular antigen testing with 50% sensitivity, for wild type and Omicron), and for W-A-2w-70 and O-A-2w-70 (regular antigen testing with 70% sensitivity, for wild type and Omicron), had $R_e$ values of less than 1 when $R_0$ = 2.5. When $R_0$ = 5.0, *Re* was greater than 1 in all the scenarios (Fig 7A). Fig 7B shows the *Re* values when additional tests were performed. When $R_0$ = 5.0 (rectangles), *Re* is larger than 1 in all scenarios, but they are less than 1 when $R_0$ = 2.5 (rectangles). In the daily testing scenario in the additional testing, the *Re* for antigen tests (0 reading time, O-A-1d-0d) is slightly lower than that for PCR tests (1 day reading time, O-P-1d-1d). Antigen testing has a lower sensitivity than PCR testing, but the advantage of a shorter reading time for results outweighs the disadvantage of lower sensitivity.

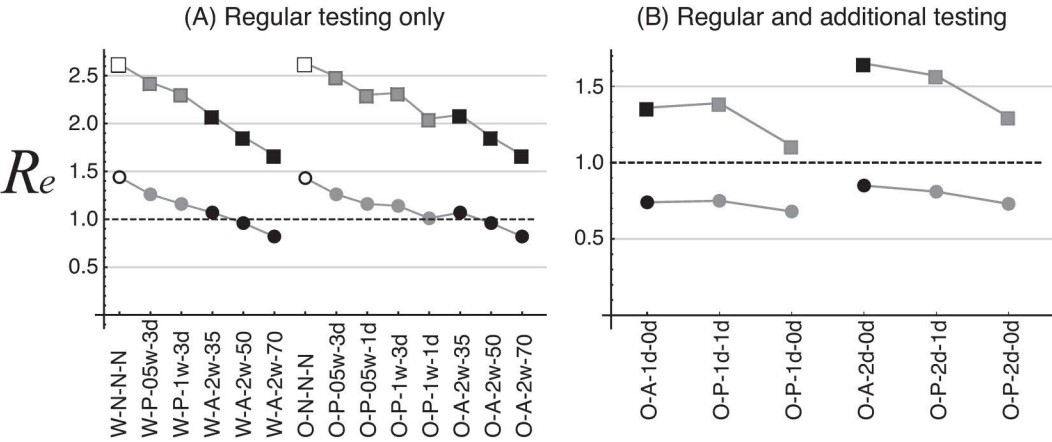

**Fig 7. Estimated effective reproductive number (*Re*) under various countermeasures.** The dots are the results for $R_0$ = 2.5 and the rectangles are the results for $R_0$ = 5.0.

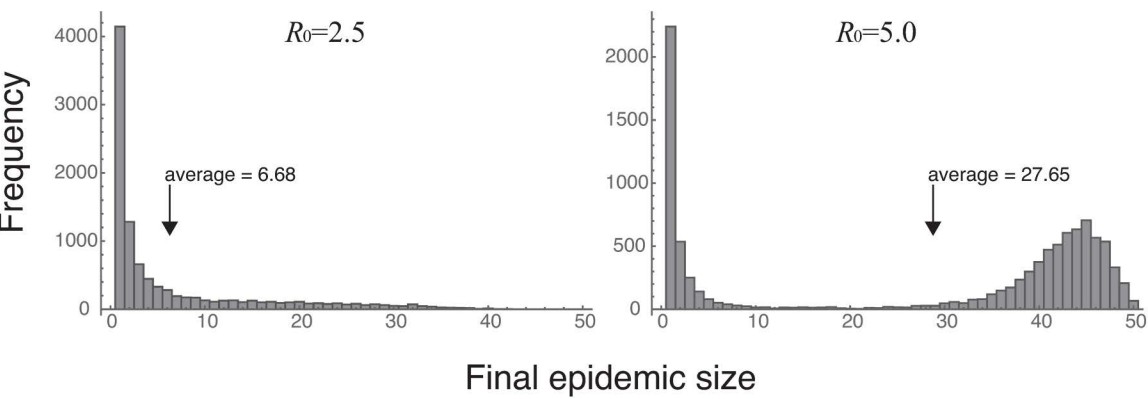

**Fig 8. The final epidemic size.**

### Cautionary note regarding average-based evaluation

In all the figures thus far, the values have been plotted as the means of 10,000 Monte-Carlo iterations. Fig 8, on the other hand, shows a histogram of the total number of infected individuals at the cessation of infection (FES) for the scenario W-P-1w-3d (regular-testing scenario for the wild type, with weekly PCR testing and a reading time of 3 days). The right panel shows the results for $R_0 = 2.5$ and the left those for $R_0 = 5.0$. When $R_0 = 2.5$, larger FES values are seen to decrease monotonically. When $R_0 = 5.0$, the frequency is the largest at the smallest FES; it decreases briefly and then rises as FES becomes larger, yielding, overall, a distribution with two peaks. The average value is, in fact, 27.65, but note that this value is rarely seen in the simulation results.

Histogram of the final epidemic size (total number of infected individuals at the cessation of infection, including the first infected individual; i.e., the number of infected individuals plus the initially infected individual) of 10,000 Monte-Carlo iterations under the testing scenario W-P-1w-3d (regular-testing scenario for the wild type with weekly testing and a reading time of 3 days). The results in the figure are based on independent runs of Monte-Carlo iterations of the results up to this point in the analysis; hence the average values shown here are slightly different from those in the previous figures and S2 Appendix.

### Discussion

Pursuing an efficient infectious-disease testing system is essential for infectious disease control. Efficiency is determined on the basis of various factors (such as basic reproduction number, the sensitivity of the test, or the cost of the test). It is important to develop a methodology that allows us to evaluate test regimes by using simulators that can flexibly incorporate these factors. We selected factors with the management of COVID-19 in mind, but if the factors are selected appropriately our model can be applied to other infectious diseases as well. Here, we investigated whether PCR tests, which have high sensitivity but are costly and difficult to perform frequently, or antigen tests, which have low sensitivity but are inexpensive and easy to perform frequently, are more cost-effective in controlling COVID-19 infections in small groups such as professional sports teams.

Fig 3A shows the numbers of infected individuals for two COVID-19 variants, the wild type and Omicron, with regular testing protocols. The number of infected individuals was less, in the case of both variants, with weekly than with 2-weekly PCR testing scenarios (Fig 3A). However, the number of such individuals did not differ greatly between the two

types of PCR testing scenario: for example, the number of infected individuals was 29.2 in scenario W-P-05w-3d (2-weekly PCR testing with a reading time of 3 days) and 26.7 in scenario W-P-1w-3d (weekly PCR testing with reading time of 3 days) for the wild type with $R_0$ = 5.0. The number of PCR tests required until all infected individuals recovered or were quarantined was much higher in scenario W-P-1w-3d (207 PCR tests) than in scenario W-P-05w-3d (120 PCR tests) (Fig 4A). In the twice-weekly antigen-testing scenario with a test sensitivity of 50% (scenario W-A-2w-50), the number of infected individuals was 19.2—less than in both PCR testing scenarios. These results indicate that, even in the case of lower sensitivity tests, more frequent testing can more effectively control the number of infected individuals.

The comparative relationship between the increased cost of a high test frequency and highly sensitive but high-cost testing (i.e., frequent antigen testing vs. delayed-reading-time PCR testing) was also analyzed (Fig 6). This information is critical for helping professional sports groups to make decisions about which testing scenario to adopt. The total cost, including not only testing but also the loss of revenue from players and staff because of their inability to be involved in practices and matches, and the cost of postponing or canceling matches owing to mass infection, was calculated. In the regular-testing scenarios, the cost of twice-weekly antigen testing was less than that of 2-weekly and weekly PCR testing (Fig 6A and 6B). The proportion of the total costs due to loss of revenue from players or staff and postponement or cancelation of matches (i.e., costs unrelated to testing) increased with increasing $R_0$ or $P_0$.

In the testing scenarios involving increased test frequency after infected individuals had been detected (i.e., the additional-testing scenarios), the total cost was less in scenarios with daily or alternate-day antigen testing than in those with PCR testing (Fig 6C and 6D). In particular, daily or alternate-day antigen testing was less expensive and more effective in the case of higher values of $P_0$ (community transmission level) and $R_0$ (basic reproductive number). When $P_0$ was low, the total cost was least in the no-testing scenario (O-N-N-N; the fine black broken line in Fig 6C and 6D); but for $P_0$ greater than $4.3 \times 10^{-4}$ with $R_0$ = 2.5 and for $P_0$ greater than $1.9 \times 10^{-4}$ with $R_0$ = 5.0, the total cost was less with the additional antigen-testing scenarios than with O-N-N-N. When the revenue loss was large (taking into account the higher incomes of professional players), the threshold value of $P_0$, which reverses the cost relationship between no-testing and additional antigen testing, decreased (see Figure A2). These results imply that twice-weekly antigen testing is more effective than weekly PCR testing, and that, when either $R_0$ or $P_0$, or both, are high, additional daily or alternate-day antigen testing is justified.

The lower total cost of the antigen vs. the PCR testing scenarios is partly due to the fact that the test results are available more quickly, thus reducing the overall number of infected individuals compared with PCR testing. In infectious disease dynamics, the effect of the reading time is important. Fig 3A compares the number of infected individuals in the regular-testing PCR scenarios for Omicron with reading times of 1 and 3 days, respectively (scenarios O-P-05w-1d and O-P-05w-3d, and O-P-1w-1d and O-P-1w-3d). For both $R_0$ = 2.5 and 5.0, the number of infected individuals in scenario O-P-05w-1d (2-weekly PCR testing with a reading time of 1 day) was less than that in scenario O-P-05w-3d (2-weekly testing with a reading time of 3 days). The difference in the number of infected individuals between these scenarios was more prominent in the scenarios with weekly PCR testing (O-P-1w-1d and O-P-1w-3d). This tendency can also be seen in the additional-testing scenarios shown in Fig 5A, where the number of infected individuals in PCR testing with a reading time of 0 days was always less than that with a reading time of 1 day. In sum, although antigen testing is less sensitive than PCR testing, the former's rapid test results outweigh the disadvantage of increased infection due

to lower sensitivity, thus keeping the number of infected individuals, the probability of mass infection, and ultimately the total cost, low.

The difference between the epidemic parameters of the wild-type and Omicron variants is in the average duration of the E state (3 days for the wild type and 1 day for Omicron). However, this difference had no impact on $R_0$ and thus no influence on the number of infected individuals under the same testing scenario (Fig 3A). Although the number of infected individuals was the same for the two variants, the probability of mass infection leading to postponement and cancelation of matches was slightly higher in the case of Omicron (Fig 3D), because the life cycle of the infection (length of time between initial infection and recovery) is shorter with Omicron (10 days) than with the wild type (12 days). This shorter life cycle reduces the number of days to the cessation of infection (Fig 3C) in the case of Omicron; and as the same number of infected individuals occurs over a shorter period, there is a greater likelihood of exceeding the threshold for mass infection (detection of five or more infected individuals within a week). This suggests that the number of infected individuals at any given time can be higher with Omicron than with the wild type, even if the final number of infected individuals is the same. It should be noted that the method of infection control and the calculation of costs may vary depending on whether the incidence or the prevalence of infection is determined to be the most important factor.

When $R_0$ was 5.0, the total number of infected individuals at the time of cessation of infection (the FES) showed a two two-peak distribution (Fig 8). The left peak, in the lower FES range, indicates that in many cases the infection had ceased, and further infection were rare. Infrequency of secondary infection is achieved through effective quarantining of infected individuals detected by testing or daily symptom checks during the early infection phase when there are few infected individuals. It is also achieved by the recovery of a very small number of infected individuals with little or no further infection. In the latter case, it is possible that the infection is not even discovered in the population. Of the 10,000 Monte-Carlo iterations (Fig 8), 1317 such cases were found when $R_0 = 2.5$ and 614 such cases were found when $R_0 = 5.0$. This means that, if $R_0 = 2.5$, no infection is discovered in the population roughly 1 in 10 times, and roughly 1 in 20 times when $R_0 = 5.0$. If the quarantining in the early phase of infection is ineffective, secondary infection rates increase, resulting in the second peak in the FES distribution, in the higher FES range when $R_0 = 5.0$. When the number of infected individuals becomes large, inadequate infection control may be observed. However, it is worth bearing in mind that, even with the same infection-control protocol, the number of infected individuals may be higher or lower than the average; the number is ultimately a matter of chance.

On the basis of the information provided by the J.League, the total number of infected individuals (including players and staff) per team was roughly 6 in the cases where infected individuals were actually detected. (As mentioned above, there may have been numerous cases where infected individuals were present but were not detected in the given group, and therefore 6 may not accurately represent the final epidemic size.) From the results of the Monte-Carlo simulations, in the case of Omicron with $R_0 = 5.0$, the number of infected individuals detected (including the first infected person) was 15–22 in the regular-testing scenarios (Fig 3) and 9–20 in the additional-testing scenarios (Fig 5). (See supporting information, S1 Table, for the average values of all measured items.) These numbers were higher than the numbers of actual cases reported by the J.League. When $R_0 = 2.5$ in the simulations, the number was 2–6. As the testing protocol differed among the various teams in the J.League, it is difficult to consider 6 as the average of any actual testing scenario, and it is therefore difficult to compare this number with those in our simulation. As a rough estimate, $R_0 = 5.0$ seems too high as a reflection of the actual J.League case, which would appear to be better reflected by an $R_0$ (or $Re$) value nearer to 2.5 or 3. As the $R_0$ of Omicron was previously estimated to be 9 [12],

the infection-control measures adopted by the J.League, such as the use of testing protocols and the avoidance of high-risk behaviors, appear to have been effective in reducing COVID-19 infection.

In this study, we used a model of infectious disease dynamics to investigate the relationship between the cost of control measures and the risk of infection among professional sports teams. Although the model was not highly realistic, and some factors were ignored for the sake of the simplicity (e.g., individuals of the teams were assumed to stay in comparative isolation with their teams, whereas in fact many go home at night to spend time with their families), the model provides insights into the infectious disease risk, testing protocols, and the cost of infection, in this context.

Our study had some limitations. First, we considered only the dynamics of infection in small populations. Small populations are characterized by a relatively short time from the start of infection to the end due to the effect of finite population size. In contrast, in large populations, it takes longer for infection to be resolved, and factors not taken into account in this study, such as the emergence of new mutants (as has occurred with COVID-19), have a large impact on the dynamics. Secondly, strategies that take into account the difference between vaccinated and unvaccinated populations are not considered. Our study assumed that the effective reproduction number was constant among individuals, but vaccination strategies can lead to differences in infection risk among individuals. It is obvious that vaccination coverage and effectiveness will affect costs. In particular, for emerging infectious diseases, there are no vaccines. The optimal strategy will change between before and after a vaccine is developed, but our study did not address this change. Even if a vaccine is developed, it takes time for it to be distributed throughout the population. Taking into account these vaccination rate dynamics, the system becomes even more complex. Thirdly, the risk of long COVID-19 and other complications, as well as the impact on mental health, which would also need to be counted as profit losses, was not considered. Fourthly, we did not take into account differences in the rate of infection between individuals or in the behavior of individuals outside team activities, because such data were not available. Future research is needed to develop more sophisticated models that integrate these variables. Fifthly, players' annual salaries and the costs of hosting games will vary depending on the country or region. Therefore, care is needed in generalizing our findings.

Despite these limitations, by setting values appropriate to the actual situation, the method we propose will provide the information needed to make decisions about infectious disease control measures.

## Supporting information

**S1 Table. A list of the average values of all the measurement items in our simulation. The average value is based on 10,000 runs.**
(XLSX)

**S1 Appendix. Total cost when revenue loss is 10 times the value assumed in the main text.**
(DOCX)

**S2 Appendix. The process of estimating the basic reproduction number from the number of infected individuals in a population of 50 individuals.**
(DOCX)

**S3 Appendix. Reproductive number under the testing scenarios.**
(DOCX)

## Acknowledgments

We thank Ms. Tomoko Irie and Mr. Hitoshi Sato for the information on infection control in the J.League. The preliminary results were first available on the website of the authors' institution: https://www.aist.go.jp/aist_j/new_research/2022/nr20220404/nr20220404.html. We have added an analysis and discussion to the preliminary results.

## Author contributions

**Conceptualization:** Michio Murakami, Seiya Imoto.

**Data curation:** Michio Murakami, Wataru Naito, Tetsuo Yasutaka.

**Formal analysis:** Masashi Kamo.

**Investigation:** Masashi Kamo.

**Methodology:** Masashi Kamo.

**Writing – original draft:** Masashi Kamo.

**Writing – review & editing:** Masashi Kamo, Michio Murakami, Wataru Naito, Tetsuo Yasutaka, Seiya Imoto.

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
