## [Decision Letter · Decision Letter 0]

17 Oct 2024

PONE-D-24-39260A comparison of COVID-19 testing strategies and costs for professional sports teams: A case study of J.League clubsPLOS ONE

Dear Dr. Kamo,

Thank you for submitting your manuscript to PLOS ONE. After careful consideration, we feel that it has merit but does not fully meet PLOS ONE’s publication criteria as it currently stands. Therefore, we invite you to submit a revised version of the manuscript that addresses the points raised during the review process.

We look forward to receiving your revised manuscript.

Kind regards,

Hani Amir Aouissi

Academic Editor

PLOS ONE

“T.Y. and W.N. received funding from the Japan Professional Football League. T.Y. and W.N. received grants or contracts from Kashima Antlers F.C. Co., Ltd., Kao Corporation, Yomiuri Giants, Tokyo Dome Co., Ltd., The Yomiuri Shimbun, Keio University, and the Tokyo Metropolitan Government. T.Y. received a grant or contract from Asahi Kasei Corporation. T.Y. and W.N. received consulting fees from the Japan Professional Basketball League, Yomiuri Giants, Nippon Professional Baseball Organization, Tokyo Yakult Swallows, and Mitsubishi Research Institute, Inc. T.Y. received consulting fees from Suntory Holdings Limited. W.N. received consulting fees from the Japan Professional Football League. M.K., M.M., T.Y., W.N., and S.I. attended the New Coronavirus Countermeasures Liaison Council, jointly established by the Nippon Professional Baseball Organization and the Japan Professional Football League, as experts without remuneration. T.Y. and W.N. were advisors to the Japan National Stadium. T.Y. and W.N. are advisors to the Japan Professional Football League.”

4. We notice that your supplementary figures are included in the manuscript file. Please remove them and upload them with the file type 'Supporting Information'. Please ensure that each Supporting Information file has a legend listed in the manuscript after the references list.

Reviewers' comments:

Reviewer's Responses to Questions

**Comments to the Author**

1. Is the manuscript technically sound, and do the data support the conclusions?

Reviewer #1: Yes

Reviewer #2: Yes

2. Has the statistical analysis been performed appropriately and rigorously? 

Reviewer #1: Yes

Reviewer #2: Yes

3. Have the authors made all data underlying the findings in their manuscript fully available?

Reviewer #1: Yes

Reviewer #2: Yes

4. Is the manuscript presented in an intelligible fashion and written in standard English?

Reviewer #1: Yes

Reviewer #2: Yes

5. Review Comments to the Author

Reviewer #1: Dear Dr. Kamo and research group members

This paper attempts to establish a mathematical model to describe the impact of COVID-19 testing methods on the spread and control of internal infections within a finite group. It also discusses the relevant cost-effectiveness relationship. I appreciate the contributions of all authors and believe This paper should be published due to its instructive value to occupational health and public health.

However, I have some concerns and believe that the paper requires further revision.

Major Revision Suggestions

1.Definition of Recovered (R): On page 4, the authors define “recovered (R)”, but it is not discussed in the subsequent model. It is important to note that newly infected and reinfected individuals often have different susceptibilities to the virus. If in this paper S=R, this should be clearly stated.

2. PCR Reading Time: On page 7, the authors state that the PCR reading time is 1 to 3 days. Meanwhile, on page 9, they claim that a wait of 0 days is unrealistic. However, in the mid-to-late and post-epidemic era, PCR results typically take only a few hours to half a day. Given that this paper will be published in a post-epidemic context, this expression may need adjustment.

3. Figure Confusion: In Figure 3A, it is unclear why the antigen testing scenarios is included in the PCR testing figure. The representations of the black circle and square do not seem as 0 value. The similar issue also appears in other figures.

4. Due to this paper will be published in the post-epidemic era the infection protection policies have changed fundamentally, the athors should descussion the meaing to the unkonw epidemic in the future and policy implications form this research.

5. Limitations: The authors should clearly state the limitations of this research.

Minor Revise suggestions

1.Author and Institution ID: The ID of authors and institution affiliations do not match; please verify for consistency.

Reviewer #2: Abstract

The abstract is clear and concise, providing a good overview of the study’s focus on infection control in professional sports teams. The use of the SEIR model is well-articulated, and the distinction between the two testing scenarios (regular and additional testing) is clear. The abstract effectively highlights the main outcomes, showing that antigen testing is more efficient than PCR testing in terms of costs and infection reduction.

Suggestions:

• The abstract could start with a brief statement on why infection control is particularly crucial for professional sports teams, such as the high level of contact and close proximity among players and staff.

• The line “The regular antigen testing scenario was more effective than the regular PCR testing scenario…” could be rephrased for clarity. For example: “Regular antigen testing was found to be more effective than PCR testing in reducing the number of infected individuals and associated costs.”

• The costs associated with infection control are briefly mentioned, but providing a bit more specificity (e.g., what types of costs were included, or their relative importance) could strengthen the understanding of the analysis.

• In line 20, change “Infection control for players and team staff is an important issue” to “Infection control among players and team staff is a critical concern.”

• In line 23, “operating costs” could be rephrased as “operational costs” for consistency with professional terminology.

• Double-check the sentence structure and flow for grammatical clarity, particularly where figures and comparisons are being made.

Introduction

• In line 43, change “the 2019 corona virus” to “The 2019 coronavirus (COVID-19)” for accuracy and clarity.

• In line 50, “quarantine asymptomatic infected individuals” could be clarified as “quarantine individuals with asymptomatic infections.”

• In line 73, instead of “frequent low-sensitivity testing is effective,” consider rephrasing to “frequent testing using lower-sensitivity methods have been shown to be effective…”

Suggestions:

1. In line 43, consider specifying the timing of the outbreak (e.g., "In response to the global outbreak of the COVID-19 pandemic in early 2020") to provide a precise reference and remind the reader of the timeline.

2. When listing current infection control measures (lines 47-49), you could expand briefly on why these measures alone may not be sufficient for professional sports settings, which helps justify the need for frequent testing protocols.

3. The phrase in line 55, "This testing protocol, however, was not adopted based on quantitative evaluation," is an important point but could be rephrased for clarity. You might say: "However, the current testing protocols have not been quantitatively evaluated to determine their effectiveness, raising questions about their actual impact on infection control."

4. In lines 64-65, when introducing antigen testing as an alternative, you could emphasize the potential for this method to balance cost and frequency more explicitly: "To achieve a balance between cost savings and infection control, antigen testing with rapid results is being considered as a feasible alternative."

5. While the introduction touches on existing studies in lines 71-74, it would be beneficial to highlight the gap more explicitly, such as: "Despite these studies, there is a lack of research specifically addressing the dynamics and economic considerations of infection control within professional sports teams."

6. In the last paragraph, the study objective could be stated more directly. For example: "This study aims to quantitatively assess the cost-effectiveness of different testing strategies for professional sports teams, focusing on the trade-offs between infection control, testing frequency, and associated costs."

Material and method

The "Materials and Methods" section is comprehensive, structured, and provides a clear methodology for the infectious disease dynamics model related to COVID-19. The use of a Monte Carlo simulation with 10,000 iterations and the application of an agent-based model add credibility to the analysis. The method for calculating the probabilities and the approach for discretizing the population dynamics shows thoroughness in modeling real-world scenarios.

Suggestions:

1. While the details are comprehensive, the section could benefit from clearer subheadings or divisions to separate different aspects of the methodology, such as "Model Setup," "Mathematical Equations," "Simulation Details," and "Testing Scenarios." This would improve readability and help readers locate specific information more quickly.

2. While the sensitivity values for the antigen test are based on PCR testing percentages, it would be helpful to include a brief explanation or rationale for these specific values to enhance transparency.

3. The reference to Table 2 is clear, but a brief summary of its contents would enhance understanding without the need to look at the table directly. Including more information in the text about the testing frequencies and reading times might help provide a quick reference for readers.

4. While some assumptions (like PCR's 0-day reading time) are mentioned, further elaboration on the implications of these assumptions or limitations (e.g., the generalizability of results beyond the J.League or other professional settings) would strengthen the section. Discussing the limitations of the data or the potential variations in real-world conditions would add depth.

5. Although the text references figures (e.g., probability calculations), directly integrating these values or their implications within the paragraph could improve the overall flow and understanding. Including short summaries of results or anticipated findings from each scenario would make the section more engaging.

6. The section uses both “individuals” and “members” interchangeably, which could cause slight confusion. Consistent terminology would ensure clarity throughout.

7. While the section provides a detailed explanation of the formulas, it may be overwhelming for some readers unfamiliar with mathematical modeling. Adding short, summarized explanations or visual aids (e.g., flowcharts) could help make the calculations more accessible.

8. The text assumes specific costs (e.g., quarantine cost per day, match postponement cost) but could benefit from a more explicit discussion of these assumptions and their implications. For example, acknowledging the variation in daily wages or stadium costs across different regions would provide context and increase the validity of the results.

9. The decision to exclude the costs associated with death or infection treatment is justified, given the low risk for healthy individuals. However, mentioning the potential impact of long COVID or other complications that could affect professional athletes would provide a more comprehensive analysis of the possible outcomes.

Discussion

While the study is thorough, there are several areas that could be strengthened:

• Although the model provides valuable insights, its applicability to real-world scenarios could be enhanced by incorporating additional variables, such as community transmission rates and individual behaviors outside of team activities.

• The impact of testing protocols on player and staff mental health, particularly in high-pressure environments like professional sports, could also be addressed. This aspect is increasingly recognized as critical in managing public health crises.

• Future research could explore the long-term implications of these testing strategies, particularly as new variants emerge and vaccination rates change.

6. PLOS authors have the option to publish the peer review history of their article (what does this mean? ). If published, this will include your full peer review and any attached files.

**Do you want your identity to be public for this peer review?** For information about this choice, including consent withdrawal, please see our Privacy Policy .

Reviewer #1: No

Reviewer #2: No

---

## [Author Response · Author response to Decision Letter 1]

15 Dec 2024

Please see Response_to_reviewers.docx

---

## [Decision Letter · Decision Letter 1]

22 Dec 2024

Comparison of COVID-19 testing strategies and costs for professional sports teams: A case study of J.League clubs

PONE-D-24-39260R1

Dear Dr. Kamo,

We’re pleased to inform you that your manuscript has been judged scientifically suitable for publication and will be formally accepted for publication once it meets all outstanding technical requirements.

Kind regards,

Etsuro Ito, Ph.D.

Academic Editor

PLOS ONE

Reviewers' comments:

Reviewer's Responses to Questions

**Comments to the Author**

1. If the authors have adequately addressed your comments raised in a previous round of review and you feel that this manuscript is now acceptable for publication, you may indicate that here to bypass the “Comments to the Author” section, enter your conflict of interest statement in the “Confidential to Editor” section, and submit your "Accept" recommendation.

Reviewer #1: All comments have been addressed

Reviewer #2: All comments have been addressed

2. Is the manuscript technically sound, and do the data support the conclusions?

Reviewer #1: Yes

Reviewer #2: Yes

3. Has the statistical analysis been performed appropriately and rigorously? 

Reviewer #1: Yes

Reviewer #2: Yes

4. Have the authors made all data underlying the findings in their manuscript fully available?

Reviewer #1: Yes

Reviewer #2: Yes

5. Is the manuscript presented in an intelligible fashion and written in standard English?

Reviewer #1: Yes

Reviewer #2: Yes

6. Review Comments to the Author

Reviewer #1: (No Response)

Reviewer #2: Thank you for your detailed and thoughtful response to my comments. I sincerely appreciate the effort you have put into providing clear and precise answers. Your explanations have greatly enhanced my understanding of the study and shed light on key aspects of your methodology and findings. I have no further concerns regarding this manuscript. Your commitment to high standards of ethics and scientific rigor is exemplary. Congratulations on this remarkable work!

7. PLOS authors have the option to publish the peer review history of their article (what does this mean? ). If published, this will include your full peer review and any attached files.

**Do you want your identity to be public for this peer review?** For information about this choice, including consent withdrawal, please see our Privacy Policy .

Reviewer #1: No

Reviewer #2: No

---

## [Editor Report · Acceptance letter]

PONE-D-24-39260R1

PLOS ONE

Dear Dr. Kamo,

I'm pleased to inform you that your manuscript has been deemed suitable for publication in PLOS ONE. Congratulations! Your manuscript is now being handed over to our production team.

Kind regards,

on behalf of

Prof. Etsuro Ito

Academic Editor

PLOS ONE